# Design and Experimental Validation of a LoRaWAN Fog Computing Based Architecture for IoT Enabled Smart Campus Applications [note 1]

**DOI:** 10.3390/s19153287

**Published:** 2019-07-26

**Authors:** Paula Fraga-Lamas, Mikel Celaya-Echarri, Peio Lopez-Iturri, Luis Castedo, Leyre Azpilicueta, Erik Aguirre, Manuel Suárez-Albela, Francisco Falcone, Tiago M. Fernández-Caramés

**Affiliations:** 1Department of Computer Engineering, Faculty of Computer Science, Centro de investigación CITIC, Universidade da Coruña, 15071 A Coruña, Spain; 2School of Engineering and Sciences, Tecnologico de Monterrey, 64849 Monterrey, NL, Mexico; 3Department of Electric, Electronic and Communication Engineering, Public University of Navarre, 31006 Pamplona, Spain; 4Institute for Smart Cities, Public University of Navarre, 31006 Pamplona, Spain

**Keywords:** IoT, smart campus, sustainability, fog computing, outdoor applications, LPWAN, LoRaWAN, 3D Ray-Launching, smart cities, Wireless Sensor Networks (WSN)

## Abstract

A smart campus is an intelligent infrastructure where smart sensors and actuators collaborate to collect information and interact with the machines, tools, and users of a university campus. As in a smart city, a smart campus represents a challenging scenario for Internet of Things (IoT) networks, especially in terms of cost, coverage, availability, latency, power consumption, and scalability. The technologies employed so far to cope with such a scenario are not yet able to manage simultaneously all the previously mentioned demanding requirements. Nevertheless, recent paradigms such as fog computing, which extends cloud computing to the edge of a network, make possible low-latency and location-aware IoT applications. Moreover, technologies such as Low-Power Wide-Area Networks (LPWANs) have emerged as a promising solution to provide low-cost and low-power consumption connectivity to nodes spread throughout a wide area. Specifically, the Long-Range Wide-Area Network (LoRaWAN) standard is one of the most recent developments, receiving attention both from industry and academia. In this article, the use of a LoRaWAN fog computing-based architecture is proposed for providing connectivity to IoT nodes deployed in a campus of the University of A Coruña (UDC), Spain. To validate the proposed system, the smart campus has been recreated realistically through an in-house developed 3D Ray-Launching radio-planning simulator that is able to take into consideration even small details, such as traffic lights, vehicles, people, buildings, urban furniture, or vegetation. The developed tool can provide accurate radio propagation estimations within the smart campus scenario in terms of coverage, capacity, and energy efficiency of the network. The results obtained with the planning simulator can then be compared with empirical measurements to assess the operating conditions and the system accuracy. Specifically, this article presents experiments that show the accurate results obtained by the planning simulator in the largest scenario ever built for it (a campus that covers an area of 26,000 m${}^{2}$), which are corroborated with empirical measurements. Then, how the tool can be used to design the deployment of LoRaWAN infrastructure for three smart campus outdoor applications is explained: a mobility pattern detection system, a smart irrigation solution, and a smart traffic-monitoring deployment. Consequently, the presented results provide guidelines to smart campus designers and developers, and for easing LoRaWAN network deployment and research in other smart campuses and large environments such as smart cities.

## 1. Introduction

A smart campus is an infrastructure similar to a smart city that makes use of Internet of Things (IoT) solutions [1,2,3,4,5,6] to connect, monitor, control, optimize, and automate the systems of a university. Today, a smart campus represents a challenging scenario for IoT networks, especially in terms cost, coverage, availability, latency, security, power consumption, and scalability.

The area covered by a campus varies substantially depending on the university, its location, the financial endowment, and the year of founding. For example, Berry College (Floyd County, Georgia, United States), is often considered the largest contiguous rural campus in the world: it covers 27,000 acres (109.26 km${}^{2}$) [7] of land. Other examples are the suburban/urban campuses of Duke University (Durham, NC, USA), which are deployed on 9350 acres (37.83 km${}^{2}$) [8], and the campus of Stanford University (Stanford, CA, USA), which covers 8180 acres (33 km${}^{2}$) [9]. Regardless of their initial surface area, it is common that campuses grow considerably as time goes by [10], hence institutions usually devise long-term sustainability plans to envision their growth in the future [11,12,13].

When a campus provides smart IoT services, it is necessary to provide communications connectivity to IoT nodes and gateways. Such a connectivity can be provided in a quite straightforward way indoors thanks to the use of popular technologies such as Wi-Fi, but, outdoors, technology selection becomes more complex, since it is not only necessary to provide good coverage and a cost-effective deployment, but also to decrease the communications energy consumption to maximize IoT node battery life.

To tackle such an issue in wide areas, a set of technologies grouped under the term Low-Power Wide Area Network (LPWAN) seem to be a good selection, since, in comparison to other previous technologies, they provide a wider area communications range and reduced energy consumption. In fact, LPWAN technologies have emerged as an enabling technology for IoT and Machine-to-Machine (M2M) communications [14] mainly due to their capabilities related to range, cost, power consumption, and capacity. Examples of such technologies are NB-IoT [15], SigFox [16], Ingenu [17], Weightless [18] or LoRaWAN [19] (a detailed comparison of these and other LPWAN technologies is given later in Section 2.2).

In the case of LoRaWAN, it is gaining momentum from both industry and academia [20,21,22]. LoRaWAN defines a communications protocol and a system architecture for the network. In addition, it uses LoRa for its physical layer [23], which is able to create long-range communications links and makes use of a Chirp Spread Spectrum (CSS) modulation that conserves the power features of Frequency Shifting Keying (FSK) while increasing its communications range. All these features make LoRaWAN a good candidate for providing wireless communications to outdoor IoT nodes in a smart campus.

Traditionally, gateways connect the IoT nodes with the cloud and among them. The cloud is basically one or more servers with large computational power, communication, and storing capabilities that receives, processes, and analyzes the data collected from the IoT nodes by performing computational-intensive tasks. Although cloud-based solutions are appropriate at a small scale, when the number of IoT nodes grows significantly and, consequently, the network traffic they generate, congestion may lead to increasing latency responses and slower data processing. Among the different alternatives to confront this challenge and to guarantee a flexible, scalable, robust, secure, and energy-efficient deployment of IoT networks, the design and implementation of a fog computing architecture was chosen. Fog computing supports physically distributed, low-latency (e.g., real-time or quasi real-time responses) and location-aware applications that decrease the network traffic and the computational load of traditional cloud computing systems by processing in the IoT nodes most of the data generated by their sensors and actuators and unburdening the higher layers from data processing [24].

Furthermore, when designing a smart campus, it is necessary to plan how LoRaWAN gateways and nodes are deployed to guarantee good IoT node coverage while minimizing the number of gateways (i.e., minimizing the smart campus communications infrastructure cost). The problem is that there are only a few examples of academic and commercial tools that create such a planning [25,26], so developers have to adapt tools previously optimized for other technologies (e.g., Wi-Fi [27]) or have to carry out tedious empirical measurements throughout the campus [28,29].

This article confronts the mentioned challenges by designing and implementing a cost-efficient, scalable, and low-power consumption LoRaWAN fog computing-based architecture for wide areas. Specifically, the system was designed with the aim of developing novel latency-sensitive IoT outdoor applications that create more sustainable and intelligent campuses. The following are the main contributions of the article, which as of writing, have not been found together in the literature:To establish the basics, it presents the main characteristics of the so-called smart campuses together with a detailed review of the state of the art of the main and the latest communications architectures and technologies, previous academic deployments, novel potential LPWAN applications and relevant tools for radio propagation modeling and planning.It thoroughly details the design, implementation, and practical evaluation of a scalable LPWAN-based communications architecture for supporting the smart campus IoT applications.The article presents the 3D modeling of a real 26,000 m${}^{2}$ campus whose LoRaWAN wireless propagation characteristics are evaluated with an in-house developed 3D Ray-launching radio-planning simulator. The results obtained by such a simulator are validated by comparing them with empirical LoRaWAN measurements obtained throughout the campus.It details how the radio-planning tool can be used to design the deployment of LoRaWAN infrastructure for three smart campus applications: a mobility pattern detection system, a smart irrigation solution, and a smart traffic-monitoring deployment. Thus, it demonstrates the usefulness of the proposed tools and methodology, which are able to provide fast guidelines to smart campus designers and developers, and that can also be used for easing LoRaWAN network deployment and research in other large environments such as smart cities.

The rest of this article is structured as follows. Section 2 reviews the state of the art on smart campuses: their characteristics, technologies, architectures, previous relevant deployments, potential applications, and the previous work on modeling and planning a smart campus. Section 3 details the architecture of the proposed system and the characteristics of the LoRaWAN testbed implementation. Section 4 describes the proposed planning simulator and the analyzed scenario. Section 5 is dedicated to the experiments. Finally, Section 6 presents the main discussion on the lessons learned from these experiences, while Section 7 is devoted to the conclusions.

## 2. Related Work

### 2.1. Characteristics of a Smart Campus

It is first important to note that in the literature, some authors use the term smart campus to refer to digital online platforms to manage learning content [30,31] or to strategies or solutions to increase the smartness of the students [32,33,34]. In this article, the term smart campus is used for referring to the hardware infrastructure and software that provides smart services and applications to the campus users (i.e., to students and to the university staff). In this regard, a smart campus, such as a smart city, can be modeled along six different smart fields [35]:Smart governance. It provides users with mechanisms to participate in decision-making or in public services.Smart people. It deals with social issues, including the engagement in campus events and learning activities.Smart mobility. This field is related to the accessibility of the campus, including the use of efficient, clean, safe, and intelligent transport means.Smart environment. It contemplates the monitoring and protection of the environment, as well as the sustainable management of the available resources.Smart living. The technologies used in these fields can monitor diverse living aspects in the campus facilities, such as personal safety [36], health [37] or crowd sensing [38].Smart economy. It is related to the competitiveness of the campus in terms of entrepreneurship, innovation, or productivity.

These smart campus fields can be further refined to determine specific smart services and solutions that should be ideally provided by a smart campus [39]:Smart living services: room occupation, classroom/lab equipment access control, health monitoring and alert services, classroom attendance systems, teaching interaction services, or context-aware applications (e.g., guidance or navigation solutions).Smart environment solutions: they include solutions for monitoring waste, water consumption, air quality (e.g., pollution) or the status of the campus green areas.Smart energy systems: they control and monitor the production, distribution, and consumption of energy in a campus.

These novel smart services and solutions make use of a growing number of enabling technologies, being the most relevant represented in Figure 1.

### 2.2. Smart Campus Communications Architectures and Technologies

In the literature, different approaches to smart campus architectures can be found, but it seems that two main paradigms drive clearly the most popular designs: IoT and cloud computing [40]. For instance, a cloud-based smart campus architecture is presented in [41]. In such a work the authors state that they were able to build their smart campus platform within three months thanks to the use of Commercial Off-The-Shelf (COTS) hardware and Microsoft Azure cloud services. Regarding IoT, it has been suggested as a tool to be considered in the architecture of a smart campus to ease the development of learning applications, access control systems, smart grids or water management systems [42,43]. Nonetheless, cloud computing and IoT solutions are often helped by Big Data techniques and Service Oriented Architecture (SOA) architectures, since they ease the processing and analysis of the collected data [44,45].

Some authors have suggested alternative paradigms for developing smart campuses. For example, in [46] a sort of opportunistic communications architecture called Floating Content is proposed that shares data through infrastructure-less services. The idea is essentially based on the ability of each Floating Content node to produce information that is shared with the interested users within a limited physical area. Other researchers propose similar architectures, but including enhancements in aspects such as security [47].

Other proposals revolve around the application of the Edge Computing paradigm and its sub-types (e.g., Mobile Edge Computing, Fog Computing), which have been previously applied to other fields [48,49]. Essentially, Edge Computing offloads the cloud from a relevant amount of processing and communications transactions, delegating such tasks to devices that are closer to the IoT nodes. In this way, such edge devices not only offload the cloud, but are also able to reduce latency response and provide location-aware services [50]. For example, in [51] the authors propose to enhance a smart campus architecture by including Edge Computing devices to provide trustworthy content caching and bandwidth allocation services to mobile users. Similarly, the authors of [52] harness street lighting to embed Edge Computing node hardware to provide different smart campus services. The Mobile Edge Computing paradigm is used in [53], where the authors present a smart campus platform called WiCloud whose servers are accessed through mobile phone base stations or wireless access points. Furthermore, other authors propose the use of fog computing nodes to improve user experience [54].

Different wireless technologies have been used to interconnect IoT nodes with smart campus platforms. For instance, BLE and ZigBee were used in [41] to provide both short and medium range communications, although ZigBee nodes can be used as relays to cover very long distances. For this latter reason, in [55] the authors make use of a ZigBee mesh network to interconnect the nodes of their campus smart grid.

Wi-Fi has also been suggested for providing connectivity [56], although the proposed applications are usually restricted to indoor locations and nearby places. Bluetooth beacons give more freedom to certain outdoor applications [57], but they require deploying dense networks that may be difficult to manage [58].

Mobile phone communications technologies (2G/3G/4G) have also been used in the literature [59], but in most cases just for the convenience of being already embedded into smartphones. 5G is currently still being tested, but some researchers have already proposed its use for providing fast communications and low-latency responses to smart campus platforms [60].

Although 5G technologies seem to have a bright future, as of writing, LPWANs are arguably one of the best alternatives for providing long-range and low-power communications. There are different LPWAN technologies such as SigFox [16], Random Phase Multiple Access (RPMA) [17], Weightless [18], NB-IoT [15], Telensa [61] or NB-Fi [62]. Among such technologies, NB-IoT, SigFox and LoRa/LoRaWAN are currently the most popular (their main characteristics are shown in Table 1).

There are several recent studies on the performance of LoRa/LoRaWAN technology for certain scenarios, but only a few describe real-world LoRaWAN deployments explicitly aimed at providing communications to a smart campus. For instance, Loriot et al. [63] conducted LoRaWAN measurements in a French campus both outdoors and indoors and showed that the technology can provide good performance over the major part of the campus. Another development is presented in [64], where the authors set up a LoRaWAN-based air quality system in their campus. Other interesting paper is [65], which details the design of a LoRa mesh network system within a campus. Finally, in [66] the authors briefly describe a smart campus platform that includes a LoRaWAN network to support faculty research projects.

### 2.3. Smart Campus Deployments

Despite the existence of many well-documented smart campus applications, there are only a few academic articles that describe in detail the deployment of real smart campuses.

For instance, in [67] an overview of the neOCampus of the Toulouse III Paul Sabatier University (France) is given. Such a smart campus runs different projects to make use of collaborative Wi-Fi, it provides an open-data platform, it fosters the reduction of the ecological footprint related to human activities and it aims for protecting the biodiversity of the campus.

Another interesting smart campus is detailed in [56], where an IoT platform deployed across different engineering schools of the Universidad Politécnica de Madrid in its Moncloa Campus (Spain) is described. Such an IoT platform is based on a cloud that provides services that follow the SOA paradigm. Two main applications are implemented: one for monitoring people flows and another for environmental monitoring.

The Sapienza smart campus (Italy) roadmap is described in [68]. Such a paper is interesting since, although it is a theoretical approach, it indicates how to structure the services to be provided by the smart campus infrastructure to scale it appropriately.

In [45] the author gives details on the Birmingham City University smart campus (United Kingdom). The smart campus platform integrates diverse business systems and smart building protocols thanks to an Enterprise Service Bus (ESB) and to the use of a SOA architecture, which provides scalability, flexibility, and service orchestration.

In the United States, an example of smart campus can be found in West Texas A&M University [66]. According to the authors, the smart campus is based on the IoT principles and covers an area of 176 acres, requiring connecting more than 42 different buildings. The described project is focused on two main tasks: to foster IoT collaboration and to provide an appropriate security framework. The proposed system has already supported diverse IoT projects, such as a LoRaWAN pilot for monitoring environmental conditions or an OpenCV-based smart parking system.

Finally, a smart campus for Wuhan University of Technology (China) is proposed in [69]. In such a paper the authors depict an architecture based on the IoT paradigm and in cloud-computing infrastructure that supports multiple applications.

### 2.4. Potential Smart Campus LPWAN Applications

Although a smart campus can support multiple indoor applications [70], in general, in such environments IoT nodes have access to power outlets and their communications can be usually easily handled with already common communications transceivers (e.g., Wi-Fi, Bluetooth, Ethernet). In contrast, this article focuses on the challenging environments that arise outdoors due the usual dependency on batteries to run IoT nodes and the need for exchanging data at relatively long distances (at least several hundred meters and up to 2 km), where LPWAN devices outperform other popular communications technologies.

The following are some of the most relevant outdoor applications that have already been implemented by using LPWAN technologies:Smart mobility and intelligent transport services. These applications require ubiquitous outdoor coverage to provide continuous data streams. For instance, in [71] researchers of Soochow University (China) propose the deployment of different smart mobility applications for their campuses, which include automatic vehicle access systems, a parking guidance service, a bus tracking system, or a bicycle rental service. Other authors also proposed similar solutions for providing campus services for smart parking [72], electric mobility [73,74], smart electric charging [75], the use of autonomous vehicles [76] or bus tracking [77].Smart energy and smart grid monitoring. Certain energy sources (e.g., renewable sources such as photo-voltaic panels or windmills) and smart grid components may be in remote locations, so it would be helpful to make use of LPWAN technologies to monitor them. For this reason, in recent years, special attention has been given to smart campus microgrids [78], smart grids [79] and smart energy systems [80].Resource consumption efficiency monitoring. These fields include waste collection [81], water management [82], energy monitoring [83], power consumption optimization [84] and sustainability [5].Campus user profiling. It is interesting for the campus managers to determine user patterns and behaviors to optimize the provided services. Thus, user profiling can be helpful to obtain mobility patterns, student daily walks, user activities, or social interactions, which can be obtained through opportunistic messaging apps [85], Wi-Fi monitoring [86] or on-board mobile phone sensors [87].Outdoor guidance and context-aware applications. This kind of systems are usually based on sensors and actuators spread throughout the campus and help people to reach their destination. There are examples in the literature of systems for guiding hearing and visually impaired people [88] or for navigating through the campus paths [89]. There are also augmented reality guidance applications [90], but it is important to note that LPWAN technologies could only help in small packet exchanges (e.g., for transmitting certain telemetry or positioning data), since the real-time multimedia content that can be demanded by augmented reality applications requires high-speed rates to preserve a good user experience.Classroom attendance. Some university events are carried out outdoors, what makes it difficult to control classroom attendance. To tackle such an issue, some researchers have proposed different sensor-based student monitoring systems that can be repurposed to be used outdoors [91].Infrastructure monitoring. It is possible to monitor remotely the status of certain assets that are scattered throughout the campus. For example, some authors presented smart campus solutions for managing campus greenhouses [92] or for monitoring high power lines with Unmanned Aerial Vehicles (UAVs) [93].Remote health monitoring. Smart campus technologies receive medical data in real time [94] or for measuring student stress [95]. Some researchers have even proposed to monitor the health of the campus trees [96].

### 2.5. Smart Campus Modeling and Planning Simulators

A smart campus is similar to an urban microenvironment where different buildings coexist with streets, open areas, parking lots, trees, benches, or people, among others. Different propagation channel models have been presented in the literature to characterize electromagnetic propagation phenomena in this type of scenarios, ranging from empirical methods to deterministic methods based on Ray Tracing (RT) or Ray-Launching (RL) approaches.

Empirical methods are based on measurements and their subsequent linear regression analysis. These methods are accurate for scenarios with the same characteristics as the real measurements, but their main disadvantages are the lack of scalability and the expensive cost in time and resources that is required to perform a measurement campaign. For example, in [97,98] empirical path loss models in a typical university campus are proposed for a frequency of 1800 MHz. The authors of [99] present a spatially consistent street-by-street path loss model for a 28 GHz channel in a micro-cell urban environment. The main drawback of these results is that the suitability of these models for path loss prediction has yet to be confirmed by the literature in other campuses.

On the other hand, deterministic methods are based on Maxwell’s equations, which provide accurate propagation predictions. These approaches usually consider the three-dimensional geometry of the environment and model all propagation phenomena in the considered scenario. Their main drawback is the required computational time, which may not be afforded in large and complex scenarios. For this reason, RT or RL techniques, which are based on Geometrical Optics (GO) and the Uniform Theory of Diffraction (UTD), have been widely used for radio propagation purposes as an accurate approximation of full-wave deterministic techniques. For example, the authors of [100] modeled the dominant propagation mechanisms using advanced RT simulations in an urban microenvironment at 2.1 GHz, showing that diffuse scattering plays a key role in urban propagation. In addition, in [101] the importance of scattering when analyzing outdoor environments in the presence of trees is reported. In [102,103] the authors present propagation analyses (at 949.2 MHz and 2162.6 MHz, respectively) that make use of RT tools in a university campus with a Universal Mobile Telecommunications System (UMTS) base station placed on rooftops. Similarly, in [104], the radio-planning analysis of a Wi-Fi network in a university campus is presented using an RT approach. In addition, it has also been reported in the literature the calibration of RT simulators for millimeter-wave propagation analyses based on the measured results in a university campus at 28 GHz [105] and 38 GHz [106]. However, all the presented works are focused on micro-cellular environments and on analyzing the wireless propagation channel between a base station at a certain height in a building and a client in a pedestrian street. In this work, a radio wave propagation analysis for the connectivity of IoT LoRaWAN-based nodes by means of an in-house developed RL algorithm is reported (which is later detailed in Section 4).

### 2.6. Key Findings

After analyzing the state of art, it is clear that a smart campus faces similar challenges to smart cities and that they share certain use cases and communications technologies. Table 2 summarizes the characteristics of the most relevant smart campuses and related solutions, and compares them with the proposed system for the University of A Coruña.

As can be observed, the systems in the table make use of diverse short-range and long-range communications technologies, multiple sensors and actuators, different hardware and software platforms, and provide services for several practical use cases. Although some systems provide a holistic approach to a smart campus, devising potential outdoor use cases and applications, their implementations are mainly focused on environmental aspects, missing other smart fields such as the ones defined in Section 2.1.

In addition, only a few solutions consider the use of fog computing, and even less made use of network planning tools. In comparison, the solution proposed in this article is one of the few academic solutions that deploys LoRaWAN infrastructure. Moreover, the proposed smart campus is almost the only one conceived from scratch to harness the benefits of fog computing.

Regarding the efficiency in the network deployment in terms of cost, coverage, and the overall energy consumption, most of the academic papers do not give any insight regarding the heterogeneous network planning, although in some cases it is specified that it is low cost (without further details). Just a couple of systems give details for the hardware used to build the demonstrator at the level that is described later in this paper in Section 3.

## 3. Design and Implementation of the Smart Campus System

### 3.1. Architecture for Outdoor Applications

The proposed communications architecture is depicted in Figure 2. As can be observed, it comprises three different layers. The layer at the bottom consists of the different IoT LoRaWAN nodes that are deployed throughout the campus. Such nodes communicate with LoRaWAN gateways that comprise the Fog Layer, since they also act as fog computing gateways, thus providing fast location-aware responses to the LoRaWAN node requests. Every fog gateway is essentially a Single Board Computer (SBC) that embeds Ethernet, Wi-Fi, and Bluetooth interfaces besides a LoRaWAN transceiver. Finally, the top layer is the cloud, where user applications run together with data storage services.

### 3.2. Operational Requirements for Outdoor Applications

To cover potential smart campus outdoor applications, a set of operational requirements grouped by capabilities was defined, including:Coverage capabilities. The coverage of the smart campus should be maximized. The typically expected coverage should be around 1 km${}^{2}$ considering both Line-of-Sight (LoS) and No-Line-of-Sight (NLoS) scenarios.Robustness capabilities. The system should provide robustness to signal interference and/or loss of network operation. The network should provide redundancy and thus be robust against single points of failure.Supported services and applications. The previously mentioned applications (in Section 2.4) should be supported. Quality of Service (QoS) requirements should include support for high-peak rate demand, latency-sensitive traffic, and location-aware IoT applications. A transmission speed of up to 50 Kbps should be expected.Deployment features and cost. It should be expected that the deployment will depend largely on low-cost IoT nodes resource-constrained in terms of memory, battery, computing capabilities, and energy consumption.Network topology. The network architecture should support Point-To-Multipoint (PMP) and Point-to-Point (PtP) links. The system should be capable of establishing ad-hoc networking for specific scenarios (i.e., by using star or mesh topologies).

### 3.3. LoRaWAN Testbed Implementation

A LoRaWAN network consisting on a gateway and several nodes was deployed. A RisingHF RHF0M301 module [108] installed on a Raspberry Pi 3 was selected as gateway. Such a module is equipped with a dual digital radio front-end interface with a typical sensitivity of −137 dBm. The module is capable of simultaneous dual-band operation and supports Adaptive Data Rate (ADR), automatically changing between LoRa spreading factors. It can use a maximum of 10 channels: 8 multi Spreading Factor (SF) channels (SF7 to SF12 with 125 KHz of bandwidth), one FSK channel and one LoRa channel. Another interesting feature of the module is its ability to operate with negative Signal-to-Noise Ratios (SNRs), with a Co-Channel Rejection (CCR) of up to 9 dB. It also supports LoRaWAN classes A, B, and C and its maximum output power is 24.5 dBm.

A 0 dBi antenna was used for the tests. The module is installed on a Raspberry Pi 3 using the provided bridge (RHF4T002). To access the LoRaWAN RFID module, configure the node, and access the message and the transmission parameters, the following software was installed on the Raspberry Pi 3: Raspbian Stretch (Linux raspberrypi 4.14.70-v7+), LoRa Gateway v5.0.1, LoRa Packet Forwarder v4.0.1 and LoRaWAN-Server v0.6.0.

The LoRa gateway was configured as follows:Coding Rate: 4/5.RX1 delay: 1.RX2 delay: 2.Power: 14 dBm.RX Frequency: 869.5 MHz.

A RisingHF RHF76-052 module [109] was used by the LoRaWAN nodes (one of such nodes can be observed on the right of Figure 3). The module has a maximum sensitivity of −139 dBm with spreading factor 12 (SF12) and 125 kHz of bandwidth, channels (0–2) at 868.1, 868.3 and 868.5 MHz, and a maximum output power of 14 dBm. During the experiments presented in this paper, the module made use of the pre-installed wired antenna.

## 4. LoRaWAN Planning Simulator Setup

### 4.1. Planning Simulator

The in-house developed 3D Ray-Launching (3D-RL) technique is based on GO and the Uniform Theory of Diffraction (UTD). The first step of such a technique is the creation of the scenario under analysis, which should consider all the obstacles within it, such as buildings, vehicles, vegetation, or people. This design phase is essential for obtaining accurate results for the real environment. Once the scenario is properly created, the frequency of operation, number of reflections, radiation pattern of the transceivers and angular and spatial resolution can be fixed as input parameters in the algorithm for simulation. Then, the whole scenario is divided into a 3D mesh of cuboids, in which all the electromagnetic phenomena are saved during simulation, emulating the electromagnetic propagation of the real waves. A detailed description of the inner-workings of the 3D-RL tool is out of the scope of this article, but the interested reader can find further information in [110].

It is worth pointing out that smart campus ecosystems are challenging environments in terms of radio propagation analysis due to their large dimensions as well as for the multipath propagation, which is caused by the multiple obstacles within them. Hence, to achieve a good trade-off between simulation computational cost and result accuracy, it is important to determine the optimal parameters for the number of reflections and the angular and spatial resolution of the RL algorithm. For such a purpose, an analysis of the optimal input parameters for the RL tool applied in large complex environments is presented in [111]. Such previous results have been considered to obtain the simulation parameters for the proposed smart campus scenario and the final values are summarized in Table 3.

### 4.2. Scenario under Analysis

This article presents a case study conducted in the northwest of Spain at the Campus of Elviña of the University of A Coruña. The campus covers an area of 26,000 m${}^{2}$ and includes elements typically found in an urban environment, such as buildings of different heights, sidewalks, roads, green areas, trees, and cars, among others. Due to the large size of the environment to be analyzed, two different scenarios were created for the simulations (in Figure 4), which correspond to the two scenarios delimited by a green and a red rectangle in the real scenario shown in Figure 5. It must be noted that the two rectangular scenarios were selected within the campus (red on the left and green on the right) to cover eight faculties. The rectangular shape is required by the 3D Ray-Launching simulator. The points marked as *M, A, Q, R* correspond to different reception distances from the transmitter (*T*). To obtain accurate results, the created scenarios include the urban elements previously mentioned. Furthermore, realistic object sizes as well as material properties (permittivity and conductivity) were taken into account. For the experiments, the LoRaWAN gateway explained in Section 3.3 was placed at the spot indicated by the red dot in the two images shown in Figure 4. Please note that the location corresponds to a single position, which is in the third floor of the faculty of Computer Science (located in the intersection of the two rectangles of Figure 5, marked in such a Figure with a blue *T*).

## 5. Experiments

### 5.1. Empirical Validation: LoRaWAN Testbed

To evaluate LoRaWAN performance in a real campus, the testbed described in Section 3.3 was deployed in the Campus of Elviña. A measurement campaign was designed and carried out to validate the chosen hardware and the simulation results provided by the 3D Ray-Launching algorithm. The performed tests consisted on transmitting packets from the LoRaWAN node to the gateway from different spots throughout the campus using acknowledgment messages. Figure 6 shows such spots (in yellow) together with the LoRaWAN gateway location (in red) for one of the two parts of the evaluated campus. The transmitter was placed near a window inside a building, at a height of 3.5 m from the street ground level. In contrast, all the measurement spots were located outdoors, at a height of 0.5 m. For every of the previously mentioned spots, the Received Signal Strength Indicator (RSSI) and Signal-to-Noise Ratio (SNR) values were recorded both at the LoRaWAN gateway and at the device that acted as a node. A Debian Virtual machine was connected to the LoRaWAN-server WebSocket endpoint. Nodes were connected through an USB port and programmed to send a 6-byte payload ten times. The firmware of the LoRaWAN node uses the USB port to create a serial interface and write the SNR and RSSI of the received acknowledgment. With this setup, the LoRaWAN node was moved to different spots and at each of them the RSSI and SNR values of the gateway and the node were recorded for a total of ten packages per location.

To test for possible interference in the used Industrial, Scientific and Medical (ISM) sub-band, a USRP B210 [112] with the same 0 dBi antenna used by the LoRaWAN gateway was connected to a laptop that acted both as data-logger and spectrum analyzer (such a measurement setup is shown in Figure 3). As an example, the result of one of the analysis during the empirical measurements is shown in Figure 7, when the system was configured to monitor a central frequency of 868.3 MHz with a sampling rate of 1 MHz.

After the measurement campaign within the campus, Radio Frequency (RF) power level estimations for the whole volume of the scenario were obtained with the aid of the 3D Ray-Launching simulation tool. The transmitter element was placed at the same position of the real LoRaWAN gateway (the red dot in Figure 6) and, using the simulation parameters shown in Table 3, a simulation was launched. The comparison between the measured RF power values and the simulation estimations is depicted in Figure 8. As can be seen in the Figure, the obtained estimations follow the tendency of the measured values, obtaining a mean error of 0.53 dB with a standard deviation of 3.39 dB (taking into account the 19 measurement points of Figure 6). The standard deviation is higher than the usual values provided by the 3D Ray-Launching. This effect could be due to size of the scenario (and the chosen simulation parameters such as cuboid size and launched ray resolution), since it is the largest scenario simulated so far by the developed 3D Ray-Launching tool. In addition, it must be noted the fact that measurements were based on RSSI values provided by the motes, which inherently add a received RF power level error. Nonetheless, the simulation results are accurate, and the simulation tool is validated satisfactorily. Regarding the results, it is worth noting the low RF power levels measured in several points of the scenario. The RF power level in many of these points is lower than −100 dBm, which is the typical ZigBee sensitivity. However, one of the advantages of the selected LoRaWAN devices is that their sensitivity is much lower (in the usual operating conditions, up to −137 dBm), as can it be observed in Table 4. Thus, the radio link budget for LoRaWAN has a higher margin, which means that longer communication distances can be achieved.

### 5.2. Planning of Smart Campus Use Cases

Once the presented 3D Ray-Launching simulation tool was validated by comparing the simulation results with the measurements, three different outdoor use cases were proposed for the smart campus environment, where LoRaWAN would have direct connectivity with a gateway:Crowdsensing/Mobility pattern detection. The purple dots depicted in Figure 9 represent the location of SBC-type devices (e.g., Raspberry Pi) that act as Bluetooth and Wi-Fi sniffers that will help to determine the mobility patterns of the users that move throughout the campus, what will optimize the deployed location-based services. In the same way, the devices could also help in crowdsensing tasks in certain areas.Smart irrigation. In this case, due to the location of the campus green areas, devices will be deployed only in one of the modeled scenarios. The device locations are represented by yellow dots shown in Figure 10. The aim of this system is to remotely control and automate the irrigation of green areas where the deployment of wired infrastructure to control the valves is very expensive or even unfeasible.Smart traffic monitoring. To detect vehicular traffic, sensors are deployed at the points represented by blue dots in Figure 11. In this way, the traffic behavior within the campus can be analyzed and the degree of parking occupancy could be inferred. Sustainability and ecological measurements to boost public transportation, to optimize routes and resources, and to adapt to real-time demand could be taken.

To validate the three mentioned smart campus use cases, 3D Ray-Launching simulations were launched for the proposed device locations. As an example, Figure 12 shows the estimated RF power distribution for bi-dimensional planes at two different heights (ground level and building’s third floor level -at the gateway’s height) for the smart irrigation use case (in Figure 12a, where the transmitter is at the center of the Green scenario) and the smart traffic-monitoring case (Figure 12b, transmitter at the furthest point of the Red Scenario). The used simulation parameters are also those shown in Table 3. As can be observed in Figure 12, the transmitter location (marked as a white circle with a T) and the morphology of the scenario (mainly the building location) greatly affect wireless signal propagation. Nevertheless, the estimated RF signal strength is quite high, taking into account the sensitivity of the employed LoRaWAN devices (i.e., −137 dBm) and the fact that the most common sensitivity value is −148 dBm (see Table 4).

To determine whether the chosen gateway location will comply with the required sensitivity for the proposed LoRaWAN node locations, 3D Ray-Launching simulation results were performed. As an example, Figure 13 summarizes the sensitivity analysis carried out for the use case illustrated in Figure 12b (i.e., for the furthest LoRaWAN node deployed for the smart traffic-monitoring use case). Specifically, Figure 13a shows the estimations obtained when the transmitter is operating at 20 dBm for different heights: ground level, third floor, and fourth floor. Figure 13b presents the same results, but for a lower transmission power (5 dBm). Finally, Figure 14 depicts the results for the sensitivity analysis based on the results obtained when transmitting at 5 dBm. This last Figure shows the areas and spots of the scenario that comply (dark blue) and do not comply (light blue) with the selected sensitivity value (in this case, the typical −148 dBm). The results show that for the case of transmitting at 20 dBm, there is no problem in terms of sensitivity threshold, but for the case of using 5 dBm, potential problems with this threshold appear within the building where the gateway is placed (this represented by the light blue surfaces at the top and left sides of the bi-dimensional planes). Therefore, a trade-off decision should be made to choose a transmission power level that ensures good sensitivity and, at the same time, the optimization of the energy consumption of the deployed motes. In fact, the results show that the gateway location could be improved by moving it from the third floor to the fourth floor. Thus, the deployment of the LoRaWAN network can be optimized by the presented 3D Ray-Launching algorithm in relation to its coverage and the overall energy consumption of the wireless communications system.

## 6. Discussion

The results presented in the previous section indicate the impact that the campus scenario has in radio-planning analysis and hence, in the determination of the optimal network layout. It must be first pointed out that the obtained results are hard to generalize, since the analyzed campus scenarios have particular characteristics that make them almost unique. Such characteristics include the size of the campus or the distribution of elements within it (mainly the buildings and their size and material properties), which have a great impact on radio signal propagation. Moreover, the topology of the deployed wireless network (i.e., the location of the nodes) has also a great influence on wireless communications performance. Therefore, the proposed methodology and solution have been validated in the presented paper, but it has to be noted that site-specific assessments are needed (that is, the results obtained for a specific campus environment cannot be generalized for any other campus scenario). Nonetheless, some aspects and results can be generalized up to a point (e.g., the received RF power for LoS situations), which are discussed in the following paragraphs.

One of the advantages in the use of LoRaWAN transceivers is their inherent low sensitivity values (in the range of −135.5 dBm to −148 dBm), which improve the reception range in comparison to other technologies. In all the observed simulation and measurement results, the received power levels are above −120 dBm, providing a sound margin to consider additional losses, such as the ones due to the variable fading related to user movements or to the dynamic elements within the campus.

In the specific outdoor applications considered for the smart campus, non-directional antennas provide adequate functionality, given the fact that theoretically, users and nodes can be located at any given location within the scenario. Nonetheless, in certain applications directional antennas may be helpful (e.g., in telemetry applications where the receiver is static, Yagi-Uda, helical or patch array antennas could be used), increasing received power levels, thus improving the communications range.

As can be observed in the experimental results, a coverage level of 20 dBm is appropriate for all the considered scenarios. However, it is desirable to reduce transmission power to reduce overall energy consumption as well as potential interference. For certain applications, tailored antennas may be considered during the network planning and the deployment phases.

The presented measurement results indicate that mainly due to the characteristics of the scenario, there is an appropriate coverage for a linear distance of 450 m with LoS (measurement point #16) and 330 m with NLoS (measurement point #19). The obtained results conclude that the location of the gateway is appropriate in terms of coverage estimation when there is LoS and in most situations where there is NLoS. The latter case requires in-depth analysis of the potential locations of the nodes, in order to consider effective losses related to building penetration, which on average can vary from 7 dB to over 25 dB depending on the building and wall structure [113].

In terms of capacity, LoRa/LoRaWAN provides a transmission speed of up to 50 Kbps, which is enough for a wide range of remote monitoring applications where users send small amounts of information or where nodes are polled with a moderate periodicity (i.e., several seconds) to provide information from sensors. Specific applications (e.g., real-time image monitoring) that require higher bandwidths can make use of alternative wireless technologies that can coexist together with the proposed LoRaWAN network.

## 7. Conclusions

After a thorough review of the state of the art on the most relevant aspects related to the deployment of smart campuses, this work detailed the design and deployment of a LoRaWAN IoT-based infrastructure able to provide novel applications in a smart campus. The architecture proposed the deployment of fog computing nodes throughout the campus to support physically distributed, low-latency, and location-aware applications that decrease the network traffic and the computational load of traditional cloud computing systems. To evaluate the proposed system, a campus of the University of A Coruña was recreated realistically through an in-house developed 3D Ray-Launching radio-planning simulator. Such a tool was tested by comparing its output with empirical measurements, showing a good accuracy. The proposed solution meets the established operational requirements:The provided coverage is roughly 1 km${}^{2}$.The system provides robustness against signal losses and interference by being able to deploy redundant gateways.The use of fog computing nodes supports low-latency and location-aware IoT applications.The maximum provided transmission speed reaches 50 Kbps.The system has been devised to make use of low-cost resource-constrained IoT nodes.The network topology support both PMP and PtP links.

In addition, three practical smart campus applications were designed and evaluated with the planning simulator (a mobility pattern detection system, a smart irrigation solution, and a smart traffic-monitoring deployment), showing that the tool is able to provide useful guidelines and insights to future smart campus designers and developers.

## Figures and Tables

**Figure 1 sensors-19-03287-f001:**
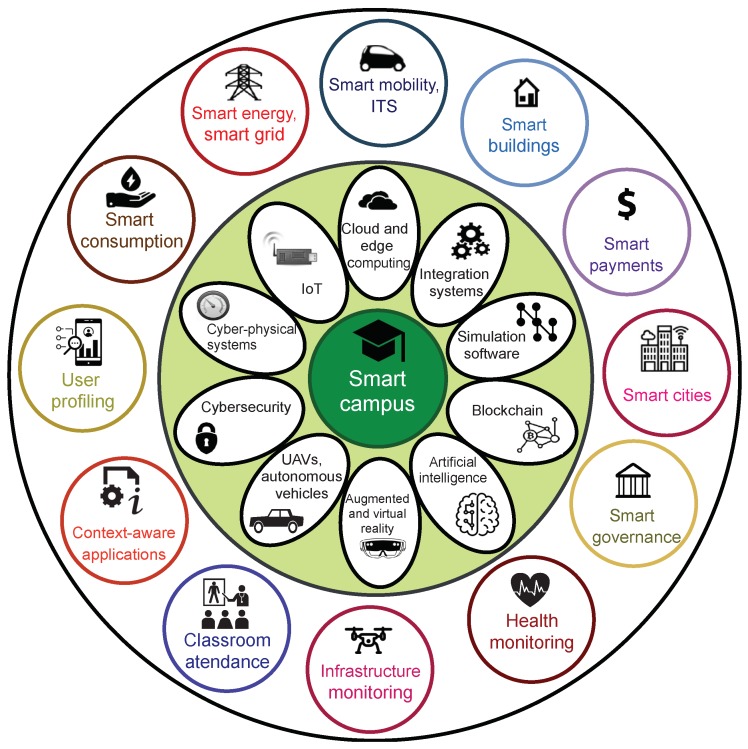
Most relevant enabling technologies and applications in a smart campus.

**Figure 2 sensors-19-03287-f002:**
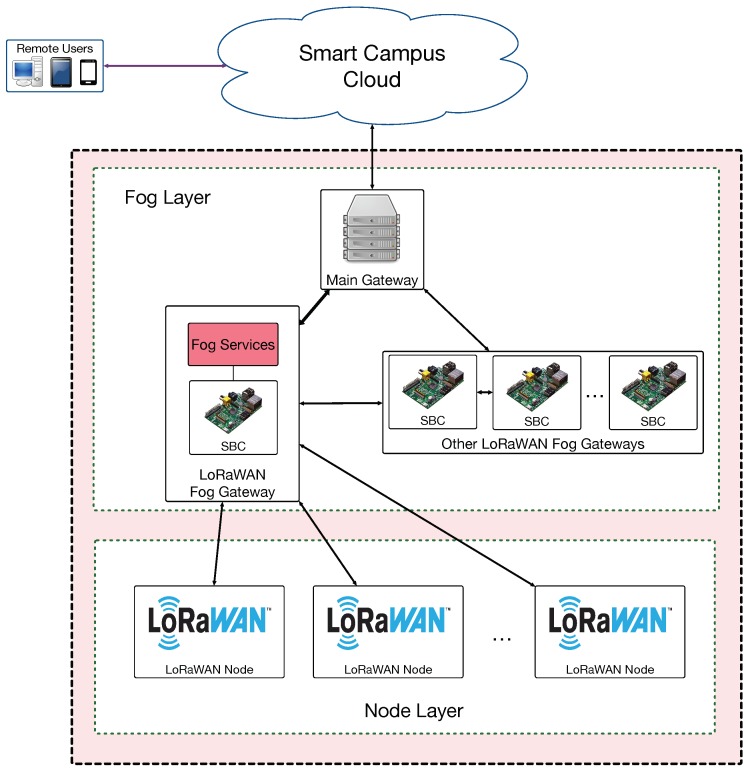
Proposed LoRaWAN-based smart campus architecture.

**Figure 3 sensors-19-03287-f003:**
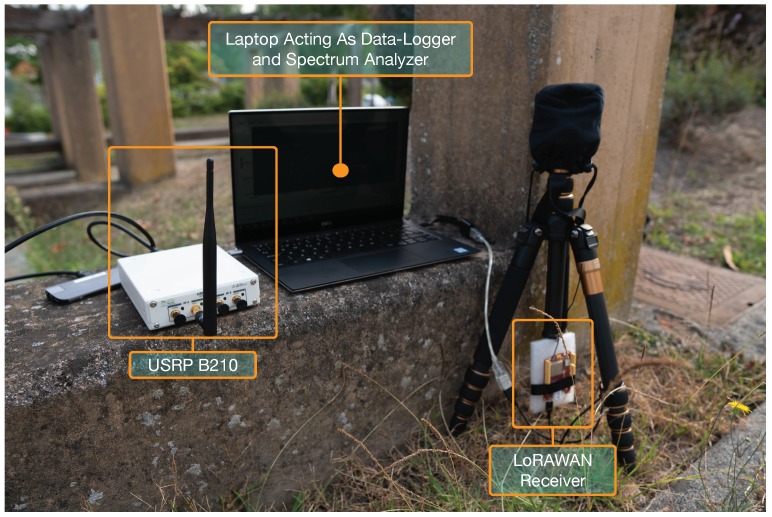
LoRaWAN IoT node during the empirical measurement campaign.

**Figure 4 sensors-19-03287-f004:**
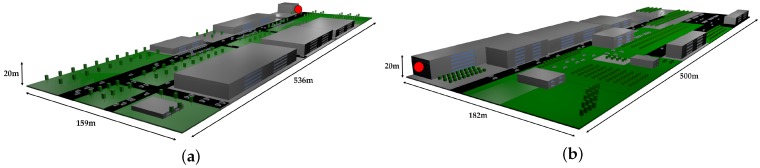
Simulated scenarios of the smart campus. (**a**) Red scenario; (**b**) Green scenario.

**Figure 5 sensors-19-03287-f005:**
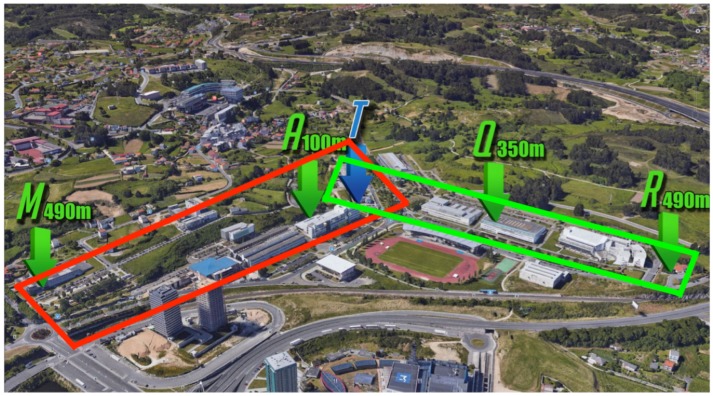
Aerial view of the Campus of Elviña, with the areas delimited for smart campus applications (Source: ©2019 Google).

**Figure 6 sensors-19-03287-f006:**
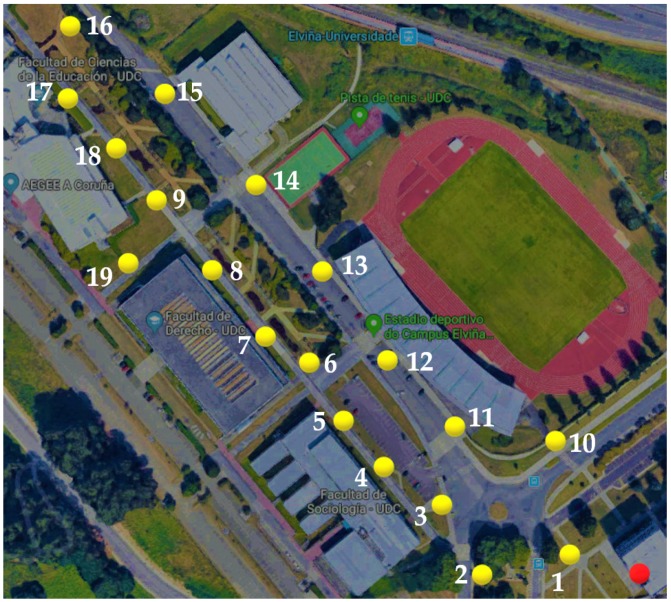
Empirical measurement points in the Green Scenario (Source: ©2019 Google).

**Figure 7 sensors-19-03287-f007:**
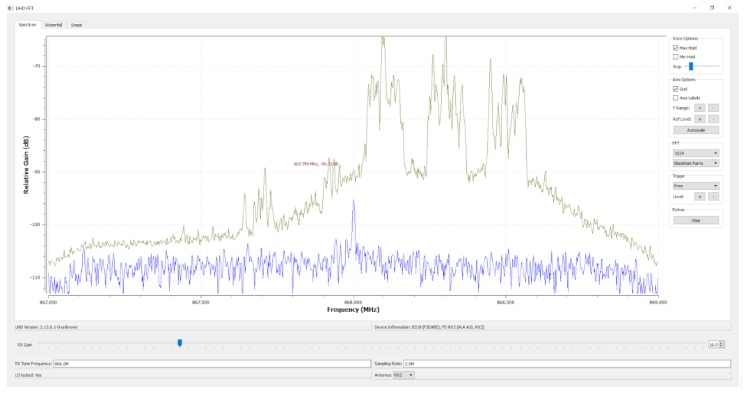
State of the radio spectrum during the performed empirical measurements.

**Figure 8 sensors-19-03287-f008:**
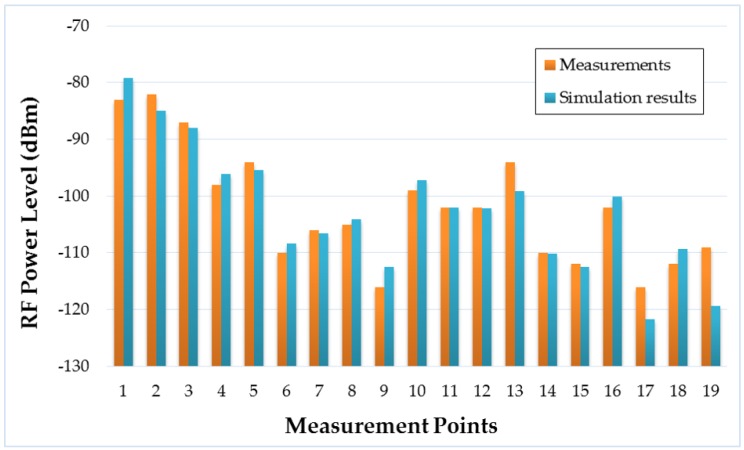
Comparison between empirical measurements and 3D-Ray-Launching simulation results.

**Figure 9 sensors-19-03287-f009:**
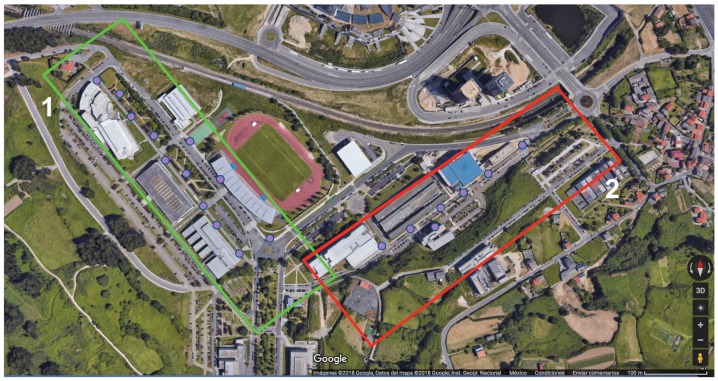
Aerial view of the spots monitored in the mobility pattern detection use case (Source: ©2019 Google).

**Figure 10 sensors-19-03287-f010:**
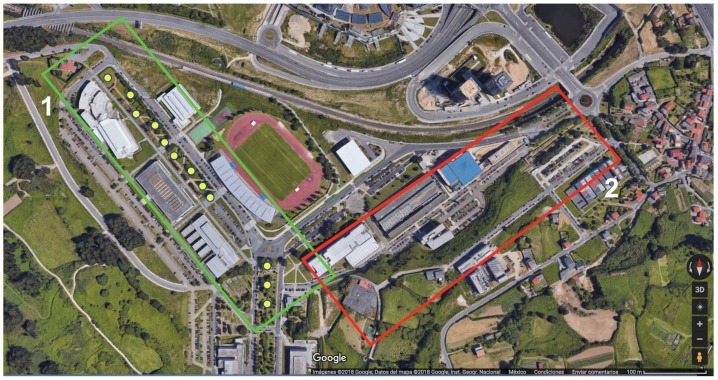
Aerial view of the smart irrigation monitoring spots (Source: ©2019 Google).

**Figure 11 sensors-19-03287-f011:**
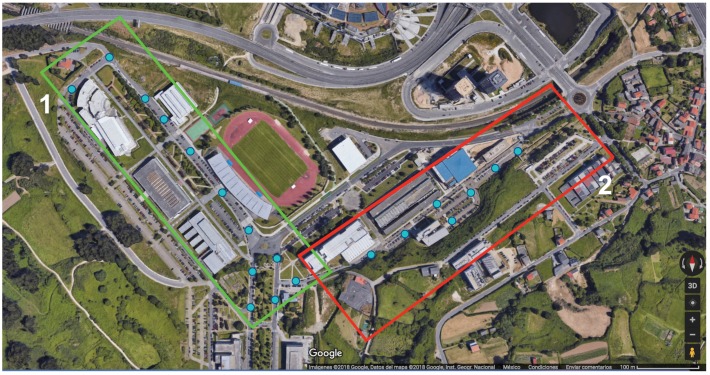
Aerial view of the spots monitored for the smart traffic use case (Source: ©2019 Google).

**Figure 12 sensors-19-03287-f012:**
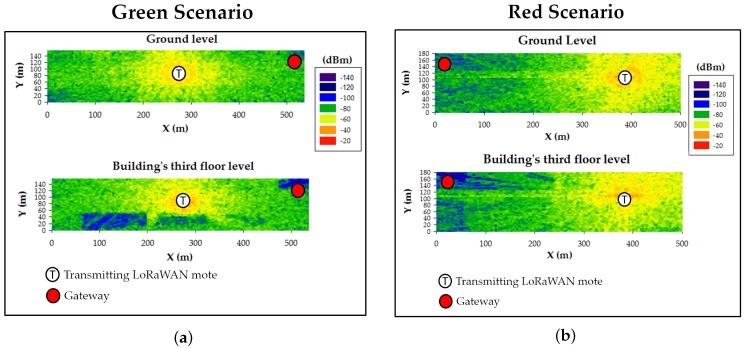
Bi-dimensional planes of the estimated RF power distribution for two different heights. (**a**) Green scenario; (**b**) Red scenario.

**Figure 13 sensors-19-03287-f013:**
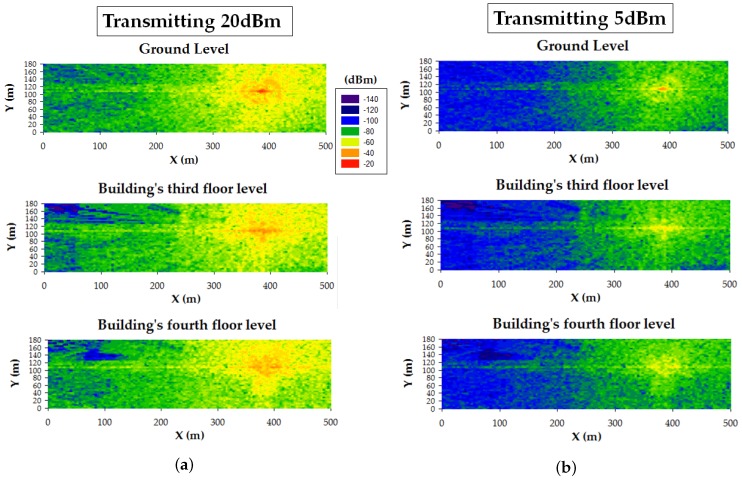
Bi-dimensional planes of the estimated RF power distribution for two different heights. (**a**) transmission power 20 dBm, (**b**) transmission power 5 dBm.

**Figure 14 sensors-19-03287-f014:**
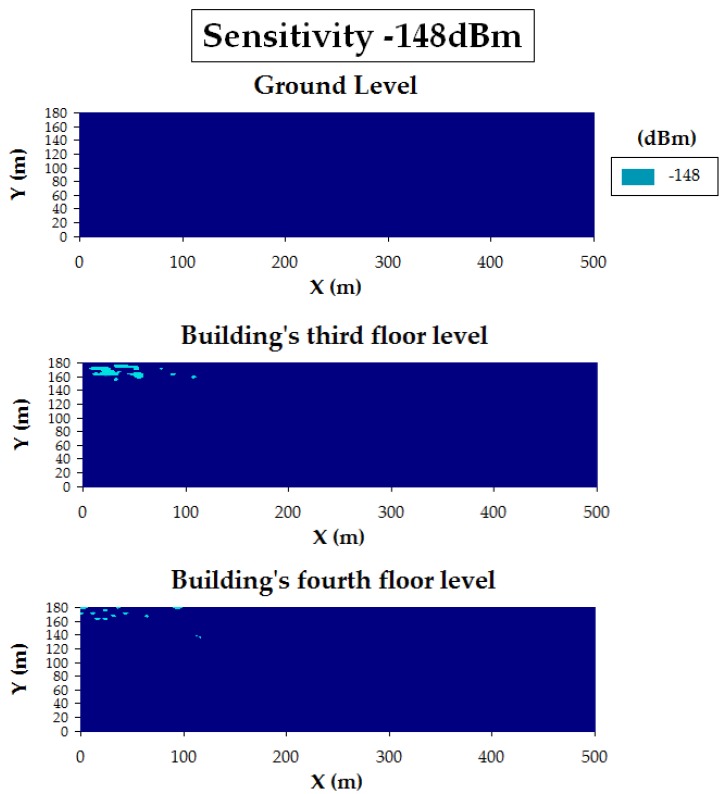
Bi-dimensional planes of the estimated RF power distribution for three different heights: sensitivity fulfillment (threshold = −148 dBm).

**Table 1 sensors-19-03287-t001:** Comparison of the three most popular LPWAN technologies.

Technology	Operating Frequency	Modulation	Maximum Range	Speed	Max. Payload	Bandwidth	Main Characteristics
NB-IoT	LTE in-band, guard-band	QPSK	<35 km	<250 kbit/s	1500 bytes	180 kHz	Low power and wide-area coverage
SigFox	868–902 MHz	DBPSK	50 km	100 kbit/s	12 bytes	0.1 kHz	Global cellular network
LoRa, LoRaWAN	Diverse UHF ISM (Industrial, Scientific, Medical) bands (e.g., 863–870 MHz and 433 MHz in Europe)	CSS	<15 km	0.25–50 kbit/s	51–222 bytes	125 kHz	Low power and wide range

**Table 2 sensors-19-03287-t002:** Comparison of the main features of the most relevant deployed smart campuses and related solutions.

Smart Campus	Area	Access Technology	Sensors and Actuators	IoT Hardware Platform	Software Platform	Use Cases	Fog Computing Capabilities	Network Planning	Sustainable Development Goals (SDGs) [107], KPIs or Results
School of STEM, University of Washington Bothell (United States) [41]	-	Zigbee, BLE, 6LowPAN	Sensor Tag 2.0 (accelerometer, magnetometer, gyroscope, light, humidity object and ambient temperature, microphone)	COTS hardware, Arduino	AWS, Microsoft Azure cloud services	-	No	No	Built in 3 months, it includes monthly cloud service bill
QA Higher Education (QAHE), University of Business and Technology, Birmingham (United Kingdom) [42]	-	-	NFC and RFID tags, QR codes	Wearables	Cisco Physical Access Control technology	Learning applications, access control systems	No	No	Deliver high quality services, protect the environment, and save costs
Tennessee State University, Nashville (United States) [43]	-	-	-	-	-	Survey on intelligent buildings, smart grid, learning environment, waste and water management and other applications	No	No	-
Northwestern Polytechnical University (China) [44]	-	Wi-Fi, Bluetooth	Built-in smartphone sensors	-	Android 2.1 platform, Big Data techniques and SOA	Where2Study, I-Sensing (participatory sensing), BlueShare (media sharing application)	No	No	-
Birmingham City University (United Kingdom) [45]	Two campuses of circa 18,000 and 24,000 m${}^{2}$, respectively	-	-	-	Microsoft’s BizTalk Server as ESB, SOA	Business systems, smart buildings	No	No	Cost savings; improved energy rating from F to B; 40% reduction in CO${}_{2}$ emissions
IMDEA Networks Institute (Spain) [46]	-	Wi-Fi, Bluetooth	-	-		Mobility model	Opportunistic Floating Content (FC) communication paradigm	No	Performance of the service in terms of content persistence, availability and efficiency
University of Oradea (Romania) [47]	-	4G, Zigbee	-	RFID labels, mobile devices, sensor equipment	Private/public cloud with steganography	No. Only architecture design	No	No	-
[51]	-	-	-	Edge computing devices	Network model and bandwidth allocation scheme for mobile users	Trustworthy content caching	Edge caching reverse auction game and bandwidth allocation for multiple contents in Mobile Social Networks	No	Resource efficiency
[52]	-	MESH Wi-Fi	Environmental sensors, IP cameras, emergency buttons	-	Neural network learning algorithms	Street lighting	Edge Computing	No	Workload prediction accuracy, resource management dashboard
WiCloud [53]	-	Wi-Fi	-	Servers, mobile phone base stations or wireless access points	Network Functions Virtualization (NFV), Software-Defined Network (SDN)	Semantic information analysis, smart class	Mobile Edge Computing paradigm	No	Historical data
WiP [54]	-	3G/4G/5G, Wi-Fi	Smartcam, smart cards, light and temperature sensor, smartphone, tablet, smartwatch	-	-	Energy consumption savings, virtual support to students, augmented reality for museum collections	Yes	No	-
Smart CEI Moncloa, Universidad Politécnica de Madrid (Spain) [56]	5.5 km${}^{2}$, 144 buildings, daily flow up to 120,000 people	Wi-Fi, Ethernet	Smart Citizen Kit (SCK)	Raspberry Pi, Arduino	Cloud, SOA paradigm	Smart emergency management and traffic restriction	No	No	Dashboard with historical data
West Texas A&M University (United States) [66]	176 acres (0.71 km${}^{2}$) campus that connects 42 buildings and a 2393 acres (9.68 km${}^{2}$) working ranch	LoRAWAN, 4G/LTE	Temperature, air pressure, relative humidity and partial concentrations	Arduino	NIST Cybersecurity Framework, standards such as COBIT and ISO	Connect cattle across the feed yard; monitor environmental conditions for network equipment; campus-wide environmental monitoring system; water irrigation; smart parking (GPS data, 800 video surveillance cameras and OpenCV-based)	No	No	-
Sapienza smart campus, University of Rome (Italy) [68]	-	N/A	N/A	N/A	Theoretical and methodological framework	Living, economy, energy, environment and mobility	No	No	Set of smart campus indicators and incidence matrix
Wuhan University of Technology (China) [69]	-	Cable, wireless, 3G/4G	Perception layer with RFID, cameras and sensors	-	Framework design, cloud computing and virtualization (Oracle 10G RAC)	Learning and living	No	No	-
Wisdom Campus, Soochow University (China) [71]	4058 acres (16.42 km${}^{2}$), 5263 staff and more than 50,000 people	-	-	-	-	Automatic vehicle access systems, parking guidance service, bus tracking system and bicycle rental service	No	No	-
IISc campus [82]	2 km × 1 km	sub-GHz radios	Low-cost ultrasonic water level sensors, solar panels	Microcontroller TI MSP432P401R	-	Water management	No	No	RSSI and Packet Error Rate (PER) performance, power budget
Ottawa City and APEC campus [104]	-	Wi-Fi	-	-	-	-	No	RT approach	Measurements and predictions of Path Loss
Universitas Indonesia [106]	Urban area	800 MHz, 2.3 GHz, and 38 GHz	-	-	RT simulators for millimeter-wave propagation analyses based on the measured results in a university campus	-	No	RT approach and physical optic near-to-far field methods	Path Loss models
**University of A Coruña (This work)**	26,000 m${}^{2}$	LoRaWAN	-	IoT nodes and SBCs (Raspberry Pi 3)	Simulations	Scalable architecture for multiple outdoor use cases	Yes	Yes (3D RL)	Planning simulator and empirical validation

**Table 3 sensors-19-03287-t003:** 3D Ray-Launching parameters.

Parameter	Value
Operation frequency	868.3 MHz
Output power level	14 dBm
Permitted reflections	6
Cuboid resolution	4 m × 4 m × 2 m
Launched ray resolution	1°
Antenna type and gain	Monopole, 0 dBi

**Table 4 sensors-19-03287-t004:** Sensitivity values for LoRaWAN devices at 868 MHz.

LoRaWAN Device	Sensitivity
Seeeduino LoRaWAN	−137 dBm
Seeeduino LoRa/GPS Shield for Arduino with LoRa BEE	−148 dBm
Dragino LoRa Shield	−148 dBm
Grove—LoRa Radio	−148 dBm
DF Robot’s LoRa MESH Radio Module	−148 dBm
Arduino MKR WAN 1300	−135.5 dBm
Adafruit RFM95W LoRa Radio Transceiver	−148 dBm
Adafruit Feather 32u4 RFM95 LoRa Radio	−148 dBm
Microchip LoRa Mote RN2483	−148 dBm
The Things Network TTN-UN-868	−148 dBm
The Things Network TTN-ND-868	−148 dBm

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
