# Peer review of "Design and Experimental Validation of a LoRaWAN Fog Computing Based Architecture for IoT Enabled Smart Campus Applications†"

_sensors, 2019, doi:10.3390/s19153287_

Reviewer 1 Report

The manuscript is extensively extended based on the conference version.

The paper is well-written and easy to follow. 

Just one suggestion, the authors could add a Discussion section to talk about the design feature, constraints, and potential improvement. 

Author Response

Dear Sir/Madam,

The authors would like to thank the reviewer for his/her valuable comments, which have certainly helped us to improve the manuscript. Please find attached our detailed responses to the comments. In order to ease the labour of the reviewers we have colored in red the major differences with the previous version of the article.

Best regards,

The Authors.

Reviewer 2 Report

This paper presents and evaluates a fog-based computing architecture for smart campuses that builds upon IoT networks and more specifically LoRaWAN technology. The paper is generally well written, and the main concepts are well explained.

While the introduction covers pretty well a variety of smart campus use cases, as well as a set of networking technologies, I would appreciate if the use of
these technologies is documented early on and not just in the related work section. This helps to create a mindset for the reader to better understand the
positioning of the work. Also, I would recommend to be more explicit about the problem statement and the key contributions of the manuscript.

The related work section describes a variety of use cases and architectures, as well as deployment cases. What I am missing somewhat is kind of a mapping of use cases and technologies from a feasibility point of view. My main concern is that these section provide various references, but little insights. In fact, one must read these respective papers to understand the bigger picture or position the paper w.r.t. these related works.

I appreciate the technical details of the design and implementation of the smart campus scenario. Although conceptually there is no significant contribution
going beyond the scientific state-of-the-art, the technical details and evaluation are interesting nonetheless. What I am missing is somewhat a critical
reflection on the obtained results. I would propose to add another discussion subsection summarizing the main results, insights and lessons learned. At this
point, the results are interesting, but hard to generalize for other deployments. For example, how specific do the authors think that the obtained
results are linked to their deployment, and observations could be generalized? What would be interesting comparitive benchmarks for other deployments? What other choices would you have made after the fact? What were your main expectations before the experiments? Were they met afterwards? Which ones were not?

The conclusion is quite short not that detailed. It says the architecture was validated. However, to derive these conclusions, one must have a set of
functional and non-functional requirements for the given use case. I would propose to highlight relevant requirements as part of section 3 (possibly a
subsection). In that respect, the conclusion section is not really exciting.

In general, I believe that the paper is interesting. It offers no significant scientific contributions beyond the state-of-the-art but aims to evaluate and
validate a specific set of technologies. However, I am missing requirements, insights, and lessons learned, which would make the paper much more valuable for a broader audience. In that sense, I would not demand for more technical work, but rather the addition of some critical reflections on the proposed work.

A minor note: please make sure that your references have all the right details.At the moment, various references are merely links to websites and online
documents, but it is not clear what are the title or authors of these documents.

Author Response

Dear Sir/Madam,

The authors would like to thank the reviewer for his/her valuable comments, which have certainly helped us to improve the manuscript. Please find attached our detailed responses to the comments. In order to ease the labour of the reviewers we have colored in red the major differences with the previous version of the article.

Best regards,

The Authors.

Round  2

Reviewer 2 Report

The authors have carefully addressed my previous comments, both in the rebuttal as well as in the revised paper. Obviously, as expected, my remarks about the generalization of the proposed solution towards other deployments could not be addressed without additional experimentation. However, I appreciate the acknowledgement of the authors that this is a limitation of the work.

Additionally, I appreciate the updated introduction, the gap with the related work and the table overview, the key requirements as well as the new discussion section. The conclusion section is now also more elaborate on the contributions of the work.  

I just have a few minor language issues which can be easily addressed in the final version of the manuscript:

-p2: low scale => small scale

-p3: It is thoroughly detailed the design => It thoroughly details the design

-p3: It is detailed how the radio planning tool => It details how the radio planning tool

-p3: Thus, it is demonstrated the => Thus, it demonstrates the

-p8:  on environmental aspects missing another smart fields => missing other smart fields (however the sentence in itself is quite unclear, what do you mean with "smart fields")

Author Response

Dear Sir/Madam,

The authors would like to thank again the reviewer for his/her valuable comments, which have certainly helped us to improve the manuscript. Please find atached our detailed responses to the comments. In order to ease the labour of the reviewer we have colored in red the major differences with the previous version of the article.

Best regards,

The authors.
